# Bisphosphonate drugs have actions in the lung and inhibit the mevalonate pathway in alveolar macrophages

Marcia A Munoz[1], Emma K Fletcher[1], Oliver P Skinner[1], Julie Jurczyluk[1], Esther Kristianto[2], Mark P Hodson[2,3], Shuting Sun[4,5], Frank H Ebetino[4], David R Croucher[1], Philip M Hansbro[6], Jacqueline R Center[1], Michael J Rogers[1]*

[1]Garvan Institute of Medical Research and St Vincent's Clinical School, UNSW Sydney, Sydney, Australia; [2]Victor Chang Cardiac Research Institute Innovation Centre, Sydney, Australia; [3]School of Pharmacy, University of Queensland, Woolloongabba, Australia; [4]BioVinc, Pasadena, United States; [5]University of Southern California, Los Angeles, United States; [6]Centre for Inflammation, Centenary Institute and University of Technology Sydney, Sydney, Australia

**Abstract** Bisphosphonates drugs target the skeleton and are used globally for the treatment of common bone disorders. Nitrogen-containing bisphosphonates act by inhibiting the mevalonate pathway in bone-resorbing osteoclasts but, surprisingly, also appear to reduce the risk of death from pneumonia. We overturn the long-held belief that these drugs act only in the skeleton and show that a fluorescently labelled bisphosphonate is internalised by alveolar macrophages and large peritoneal macrophages in vivo. Furthermore, a single dose of a nitrogen-containing bisphosphonate (zole-dronic acid) in mice was sufficient to inhibit the mevalonate pathway in tissue-resident macrophages, causing the build-up of a mevalonate metabolite and preventing protein prenylation. Importantly, one dose of bisphosphonate enhanced the immune response to bacterial endotoxin in the lung and increased the level of cytokines and chemokines in bronchoalveolar fluid. These studies suggest that bisphosphonates, as well as preventing bone loss, may boost immune responses to infection in the lung and provide a mechanistic basis to fully examine the potential of bisphosphonates to help combat respiratory infections that cause pneumonia.

*For correspondence:
m.rogers@garvan.org.au

## Editor's evaluation

Your paper is a careful analysis to understand the effects of bisphosphonates, long thought to act only on osteoclasts to block bone resorption, on the lung. You show that drugs in this class act on pulmonary macrophages and block protein prenylation. This leads to an enhanced response to bacterial endotoxin and immune response to bacterial endotoxin with increased levels of cytokines and chemokines in bronchoalveolar fluid – a series of events that may explain reduced rates of pneumonia seen in patients treated with these drugs. This clinical observation has long defied our understanding.

## Introduction

Nitrogen-containing bisphosphonates (N-BPs) are a class of bone-seeking drugs used worldwide as the frontline treatment for disorders of excessive bone resorption such as post-menopausal oste-oporosis and cancer-associated bone disease (*Russell, 2011*). By virtue of their avidity for calcium ions, N-BPs bind rapidly to the skeleton, where they are internalised by bone-degrading osteoclasts

(*Russell et al., 2008*; *Rogers et al., 2020*). Intracellularly, N-BPs disable osteoclast function by inhibiting the enzyme farnesyl diphosphate (FPP) synthase in the mevalonate pathway (*van Beek et al., 1999*; *Dunford et al., 2001*), thereby preventing the post-translational prenylation of small GTPase proteins necessary for osteoclast function (*Luckman et al., 1998*; *Fisher et al., 1999*).

There is increasing evidence that N-BP drugs, such as zoledronic acid (ZOL), have benefits beyond preventing bone loss (*Center et al., 2020*), and, unexpectedly, N-BP therapy has recently been linked to reduced risk of mortality from pneumonia (*Colón-Emeric et al., 2010*; *Sing et al., 2020*). In a randomised, controlled trial of >2000 hip fracture patients, ZOL therapy reduced the risk of death by 28% compared to placebo infusion (*Lyles et al., 2007*). Retrospective analysis also suggested that ZOL-treated patients were less likely to die from pneumonia than placebo-treated subjects (*Colón-Emeric et al., 2010*). Recently, a 'real-world' population-based, observational study of hip fracture patients aged 50 years or above also showed a significant reduction in risk of pneumonia and pneumonia mortality in hip fracture patients that had received N-BP therapy compared to no treatment or other osteoporosis medications (*Sing et al., 2020*). Similar findings were reported in the post hoc analysis of a randomised controlled trial of ZOL in women over the age of 65 years (*Reid et al., 2021*). Pneumonia is the most frequent cause of admission to ICU, and a study of long-term patients in respiratory ICU revealed a significant reduction in mortality in people treated with the N-BP pamidronate compared to those without treatment (*Schulman et al., 2016*). Furthermore, in a retrospective cohort study of ICU subjects, we showed a 59% reduction in mortality in patients treated with bisphosphonate prior to hospitalisation (*Lee et al., 2016*). However, the mechanisms underlying the surprising beneficial effects of these drugs on pneumonia and in ICU patients are unknown.

Globally, respiratory diseases constitute the most common cause of death, and thus, therapies that boost the immune response to common lung infections are urgently needed. Bacterial infections such as *Streptococcus pneumoniae*, *Haemophilus influenzae*, *Chlamydia pneumoniae,* and *Staphylococcus aureus* are the main cause of community-acquired pneumonia. Importantly, these pathogens also underlie severe complications of viral respiratory disease that can significantly increase morbidity and mortality. For example, influenza-related mortality is often associated with pneumonia caused by co- or secondary bacterial infection (*Morris et al., 2017*). In this study, we debunk the long-held view that N-BP drugs act only in the skeleton in healthy mice and show that even a single dose of N-BP in mice is sufficient to affect tissue-resident macrophages, including lung alveolar macrophages (AMΦ), boosting their response to bacterial endotoxin.

## Results and discussion
### Systemically administered N-BP is internalised by tissue-resident macrophages outside the skeleton

We previously reported that cultured macrophages and tumour-associated macrophages in vivo, like osteoclasts, have the ability to internalise N-BP by endocytosis (*Thompson et al., 2006*; *Junankar et al., 2015*). Given the important role of AMΦ in lung homeostasis and the initial immune response to respiratory infection (*Byrne et al., 2015*; *Crane et al., 2018*), we focused on whether these cells are capable of internalising a systemically administered, fluorescently labelled analogue of ZOL (AF647-ZOL). Like most other fluorescently labelled N-BPs (*Roelofs et al., 2010*), AF647-ZOL did not inhibit protein prenylation compared to the parent N-BP (ZOL) when tested in cultures of bone marrow-derived macrophages (BMDM; *Figure 1a*), but can be used to track drug uptake by cells in vivo (*Junankar et al., 2015*). Animals were injected with a single intravenous (i.v.) dose of AF647-ZOL, and cells collected by bronchoalveolar lavage (BAL) were analysed by flow cytometry. In the absence of immune challenge in mice, approximately 90% of cells in BAL were AMΦ (*Figure 1—figure supplement 1*). Importantly, >98% of AMΦ (TCRβ⁻B220⁻CD11b$^{lo/-}$CD11c$^{hi}$F4/80⁺) in BAL samples were clearly labelled with AF647-ZOL 4 hr after i.v. administration (*Figure 1b–d*).

As a comparison with AMΦ, we also examined N-BP uptake in peritoneal macrophages (PMΦ). Under baseline conditions, 80% of cells obtained by peritoneal lavage (PL) consisted of PMΦ (B220⁻ TCRb⁻ Siglec-F⁻ Ly6G⁻ CD11b⁺ F4/80⁺), most of which were CD11b$^{hi}$F4/80$^{hi}$ large PMΦ, with a less abundant population of CD11b⁺F4/80$^{int}$ small PMΦ (*Ghosn et al., 2010*; *Cassado Ados et al., 2015*; *Figure 1e*, *Figure 1—figure supplement 1*). Similar to BAL cells, approximately 80% of peritoneal cells incorporated AF647-ZOL after a single i.v. dose (*Figure 1f*), the majority of which (99%) were

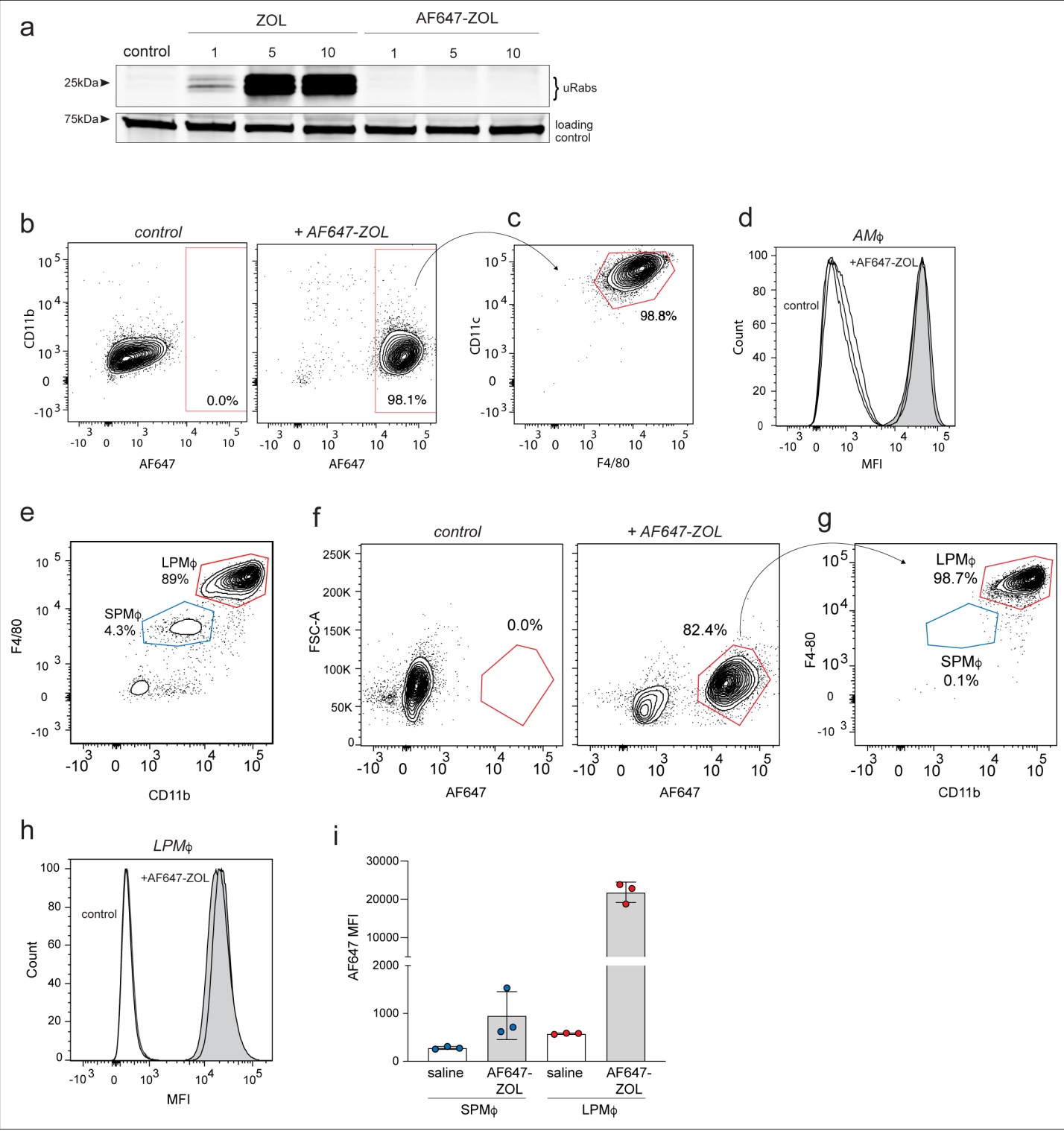

**Figure 1.** Bisphosphonate is internalised by alveolar and peritoneal macrophages in vivo. (**a**) An in vitro prenylation assay detects unprenylated Rab GTPases (uRabs) in bone marrow-derived macrophages after culture for 24 hr with 1, 5, or 10 µM zoledronic acid (ZOL) compared to 1, 5, or 10 µM AF647-ZOL. (**b**) FACS plots showing the percentage of labelled single cells in bronchoalveolar lavage (BAL) samples after one intravenous (i.v.) dose of AF647-ZOL compared to saline-treated mice. (**c**) AF647-ZOL-positive cells were predominantly alveolar macrophages (AMΦ), that is, B220⁻TCRb⁻, CD11c⁺F4/80⁺ singlets. (**d**) Histograms show AF647-ZOL mean fluorescence intensity (MFI) of AMΦ in BAL samples from n = 3 control mice (white) and n = 3 AF647-ZOL-treated mice (grey). (**e**) FACS plot illustrating the percentages of small peritoneal macrophages/SPMΦ (B220⁻TCRb⁻ singlets, CD11bᶦⁿᵗF4/80ᶦⁿᵗ) and large peritoneal macrophages/LPMΦ (B220⁻TCRb⁻ singlets, CD11bʰⁱF4/80ʰⁱ) in peritoneal lavage. (**f**) Percentage of labelled

*Figure 1 continued on next page*

*Figure 1 continued*

peritoneal cells 4 hr after one i.v. injection of saline (left) or AF647-ZOL (right). (**g**) The labelled cell population (AF647-ZOL⁺, 82.4%) in (**f**) consists predominantly of CD11b$^{hi}$F4/80$^{hi}$ LPMΦ. (**h**) Histograms show the MFI of LPMΦ from saline- (white) and AF647-ZOL-treated (grey) mice. (**i**) MFI (AF647 MFI) values from SPMΦ and LPMΦ isolated from saline- or AF647-ZOL-treated animals. Bars represent mean ± SD (n = 3 mice per group in (**h, i**); each symbol represents the measurement from an individual mouse). FACS plots in (**b, c, e–g**) are representative of three mice per group.

The online version of this article includes the following source data and figure supplement(s) for figure 1:

**Source data 1.** In vitro prenylation assay of BMDM following treatment with ZOL or AF647-ZOL.

**Source data 2.** In vitro prenylation assay of BMDM following treatment with ZOL or AF647-ZOL, showing cropped regions of the blot (uRabs and loading control).

**Figure supplement 1.** Percentage of macrophages in bronchoalveolar lavage (BAL) and peritoneal lavage (PL) samples from mice.

CD11b$^{hi}$F4/80$^{hi}$ large PMΦ (*Figure 1g and h*). In contrast, the CD11b⁺F4/80$^{int}$ small PMΦ incorporated negligible amounts of fluorescently labelled ZOL (*Figure 1i*). The reason why uptake of N-BP occurred selectively in large PMΦ rather than in small PMΦ remains to be determined, although these subsets differ in their origin, phenotype, and function (*Cassado Ados et al., 2015*).

In addition to the well-described uptake of N-BP by osteoclasts in bone (*Rogers et al., 2020*; *Coxon et al., 2008*), these findings clearly demonstrate that N-BP can also be efficiently internalised in vivo by tissue-resident macrophages outside the skeleton, including AMΦ in the lung and LPMΦ in the peritoneal cavity.

## A single i.v. dose of N-BP is sufficient to inhibit the mevalonate pathway in alveolar and peritoneal macrophages

Despite the rapid bone-targeting property of N-BPs, it has been shown in rodents that small amounts of ZOL can accumulate in soft tissues (*Green and Rogers, 2002*); however, the concentrations that can be achieved are unknown. To determine whether tissue-resident macrophages can incorporate sufficient N-BP in vivo to have a pharmacologic effect, we analysed two biochemical outcomes that, together, are reliable features of intracellular N-BP action in cells (*Rogers et al., 2020*): (i) the cytoplasmic build-up of the upstream metabolite isopentenyl diphosphate (IPP) and its isomer dimethylallyl diphosphate (DMAPP) (*Räikkönen et al., 2009*); and (ii) reduced production of the isoprenoid lipid geranylgeranyl diphosphate (GGPP), with the consequent accumulation of unprenylated small GTPase proteins including those of the Rab and Rho superfamilies (*Luckman et al., 1998*; *Figure 2a*). To address whether the potent N-BP ZOL has pharmacological effects on AMΦ and PMΦ, we used liquid chromatography tandem mass spectrometry (LC-MS/MS) to examine the accumulation of IPP/DMAPP and a sensitive biochemical in vitro assay to detect changes in the level of unprenylated Rab proteins (*Ali et al., 2015*; *Rogers et al., 2020*).

LC-MS/MS analysis showed that IPP/DMAPP was undetectable in extracts of BAL or PL cells from saline-treated mice, but there was a clear increase in the level of IPP/DMAPP in BAL and PL cells collected 48 hr after a single i.v. injection of ZOL (*Figure 2b*). Importantly, ZOL treatment also resulted in a marked accumulation of unprenylated Rab proteins in BAL and PL cell samples (*Figure 2c*). It is unlikely that such an effect of N-BP on protein prenylation in macrophages in vivo could have been detected using the relatively insensitive western blot approach previously employed to study bone-resorbing osteoclasts, which engulf large amounts of N-BP (*Frith et al., 2001*; *Rogers et al., 2020*). However, the development of a much more sensitive in vitro prenylation assay (*Ali et al., 2015*) now allows the detection of subtle effects on protein prenylation in cells outside the skeleton that may internalise much smaller quantities of N-BP.

Together with the evidence for uptake of N-BP by AMΦ and large PMΦ and (*Figure 1*), these findings demonstrate unequivocally that systemic administration of N-BP has pharmacological activity outside the skeleton in healthy mice. We show that a single dose of ZOL is sufficient to inhibit the mevalonate pathway in tissue-resident macrophages (AMΦ and PMΦ), causing a build-up of IPP/DMAPP metabolites and an accumulation of unprenylated small GTPase proteins – characteristic hallmarks of N-BP action (*Rogers et al., 2020*). Although still to be confirmed in humans, our findings overturn the long-standing textbook paradigm that N-BP drugs, which have been in clinical use for several decades (*Russell, 2011*), act only in the skeleton. Our findings also shed new light on the commonest side effect of N-BP therapy, a transient acute phase response. This was previously thought

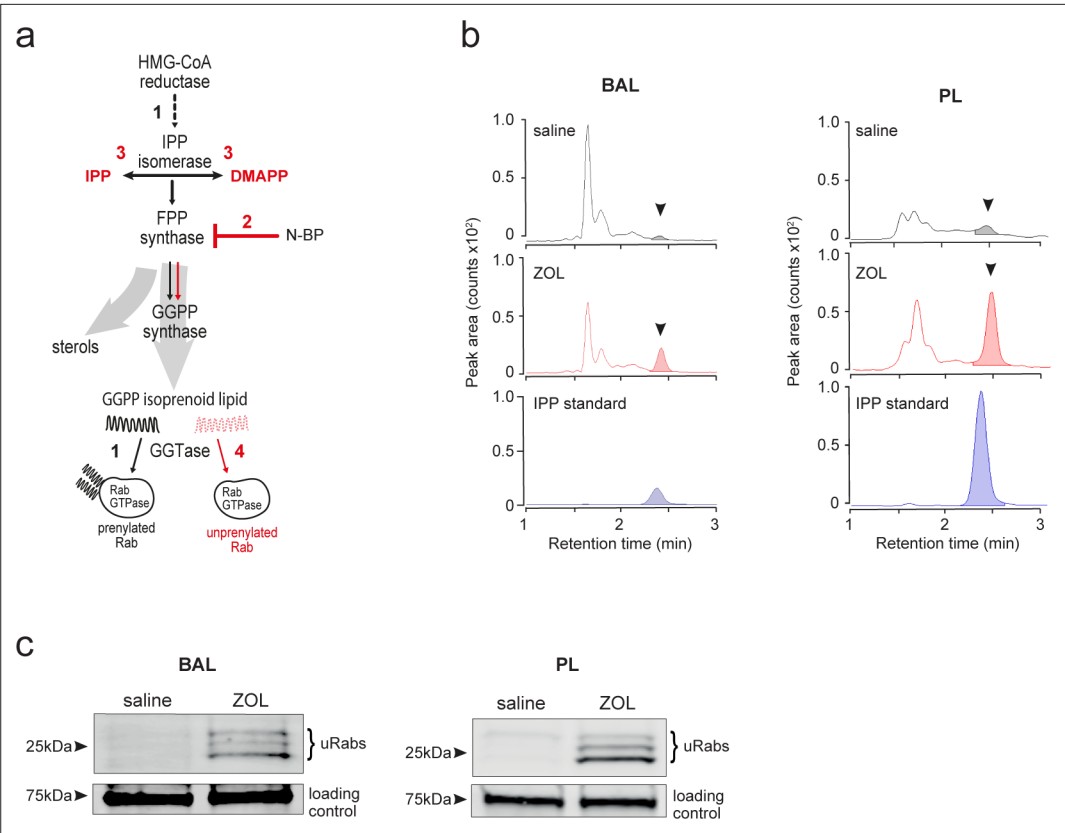

**Figure 2.** Systemically administered bisphosphonate has pharmacological activity on the mevalonate pathway in alveolar and peritoneal macrophages in vivo. (**a**) Flux though the mevalonate pathway (black arrows) enables prenylation of Rab GTPases by utilising geranylgeranyl diphosphate (GGPP) (step 1); inhibition of farnesyl diphosphate (FPP) synthase by nitrogen-containing bisphosphonate (N-BP) (step 2) causes upstream build-up of isopentenyl diphosphate (IPP) and dimethylallyl diphosphate (DMAPP) (step 3) and prevents downstream Rab prenylation by reducing GGPP synthesis (red arrows, step 4). (**b**) Detection of IPP/DMAPP (arrowhead) by liquid chromatography tandem mass spectrometry (LC-MS/MS) in cells from bronchoalveolar lavage (BAL) (left) and peritoneal lavage (PL) (right) samples 48 hr after intravenous (i.v.) zoledronic acid (ZOL) or saline treatment. Coloured peaks in the chromatogram depict the relative abundance of IPP/DMAPP. (**c**) Detection of unprenylated Rab GTPases (uRabs) in cells from BAL (left) and PL (right) samples 48 hr after i.v. ZOL or saline treatment. Data are representative of three separate experiments.

The online version of this article includes the following source data for figure 2:

**Source data 1.** Original blots of the in vitro prenylation assay of BAL cells and PL cells, following treatment of mice with a single i.v. dose of ZOL.

**Source data 2.** Original blots of the in vitro prenylation assay of BAL cells and PL cells following treatment of mice with a single i.v. dose of ZOL, showing cropped regions of the blots (uRabs and loading control) and molecular mass markers.

to be caused by the accumulation of IPP in circulating monocytes, which then activates Vγ9Vδ2-T cells to produce TNFα and IFNγ (*Roelofs et al., 2009*). Our findings suggest that tissue-resident macrophages, such as AMΦ and large PMΦ, are in fact a more likely source of the IPP responsible for the activation of Vγ9Vδ2-T cells.

## Treatment with N-BP in vivo enhances the production of cytokines and chemokines in response to immune challenge

We recently reported that loss of protein prenylation in cultured monocytes promotes the formation of the NLRP3 inflammasome, resulting in increased caspase-1-mediated processing of pro-IL-1β following bacterial endotoxin (lipopolysaccharide [LPS]) stimulation (*Skinner et al., 2019*). Therefore, we next examined whether N-BP-mediated inhibition of protein prenylation in tissue-resident

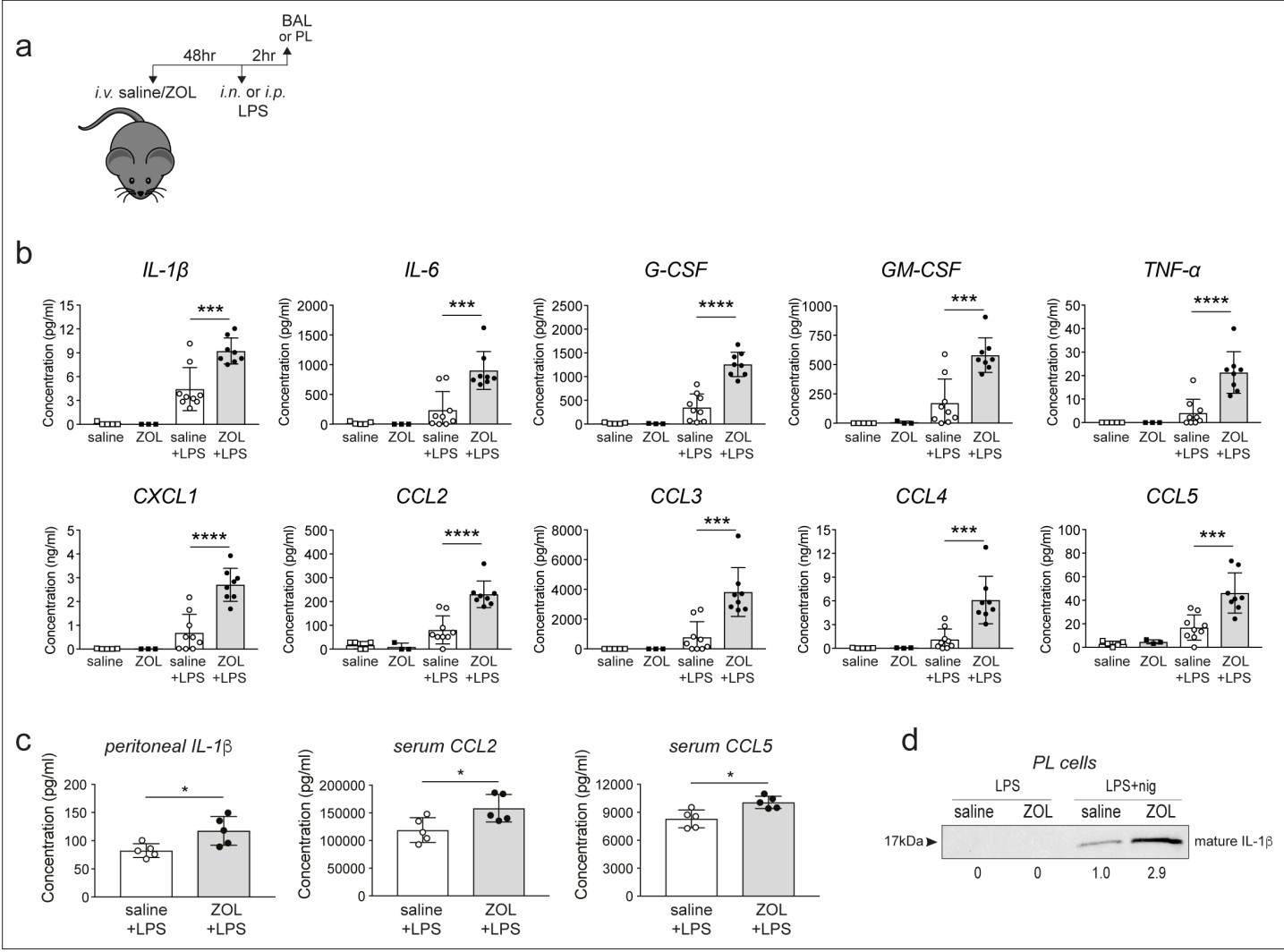

**Figure 3.** Zoledronic acid (ZOL) treatment enhances the production of inflammatory cytokines in response to endotoxin challenge in vivo. (**a**) Schedule of intravenous (i.v.) ZOL administration 48 hr prior to intranasal (i.n.) or intraperitoneal (i.p.) lipopolysaccharide (LPS) treatment and subsequent collection of bronchoalveolar lavage (BAL) or peritoneal lavage (PL) fluid, respectively. (**b**) Multiplex analysis of cytokines and chemokines in BAL fluid from saline- or ZOL-treated mice after i.n. LPS challenge, and (**c**) in peritoneal fluid, and serum after i.p. LPS challenge. In (**b**), bars represent mean ± SD, n = 8–9 mice per group with LPS, or n = 3–5 mice per group with ZOL/saline alone; ***p<0.001, ****p<0.0001, ANOVA with Tukey's post hoc test. In (**c**), bars represent mean ± SD, n = 5 mice per group; *p<0.05, unpaired *t*-test with Welch's correction; each symbol represents the measurement from an individual mouse. (**d**) Western blot detection of mature, extracellular IL-β in conditioned medium from PL cells, isolated from a ZOL- or saline-treated mouse, then stimulated ex vivo with LPS or LPS+ nigericin. Relative levels of IL-β were calculated by densitometry and are shown below each lane. The blot shown is representative of three independent experiments.

The online version of this article includes the following source data and figure supplement(s) for figure 3:

**Source data 1.** Western blot of conditioned media using anti-IL-1β shows a single 17 kDa band of cleaved IL-1β.

**Figure supplement 1.** Zoledronic acid (ZOL) treatment does not affect the percentage or number of alveolar macrophages or cell viability in vivo.

macrophages alters the response to LPS in vivo, particularly in the lung. Mice were challenged intranasally (i.n.) or intraperitoneally (i.p.) with LPS 48 hr after i.v. administration of ZOL or saline (*Figure 3a*). ZOL treatment alone did not alter the percentage, total number or viability of AMΦ recovered in BAL samples (*Figure 3—figure supplement 1*), nor had any effect on the levels of cytokines/chemokines in BAL fluid (*Figure 3b*). However, i.n. LPS administration in ZOL-treated mice resulted in a significant increase (2.5- to 5.0-fold) in the production of proinflammatory cytokines IL-1β, IL-6, TNFα, G-CSF, GM-CSF and chemokines CXCL1, CCL2, CCL3, CCL4, and CCL5 in BAL fluid compared to control mice (*Figure 3b*). This increase in cytokine and chemokine release was not associated with changes in

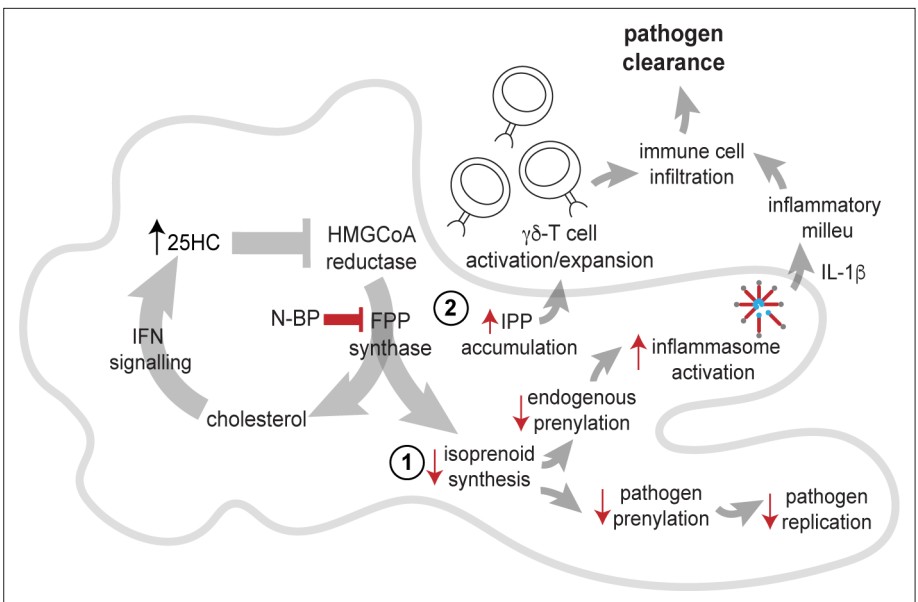

**Figure 4.** Potential routes of antimicrobial activity of nitrogen-containing bisphosphonate (N-BP) via effects on the mevalonate pathway in alveolar macrophages. Inhibition of farnesyl diphosphate (FPP) synthase by N-BP prevents the biosynthesis of isoprenoid lipids required for normal protein prenylation (step 1). Lack of prenylation leads to enhanced inflammasome activation and increased IL-1β release in response to bacterial endotoxin, boosting the initial inflammatory response and pathogen clearance. Lack of isoprenoid biosynthesis may also hinder the propagation of intracellular pathogens that depend on the host cells' mevalonate pathway. Inhibition of FPP synthase by N-BP causes the accumulation of isopentenyl diphosphate (IPP)/dimethylallyl diphosphate (DMAPP) phosphoantigens (step 2) capable of triggering activation and proliferation of human Vγ9VδT cells with antimicrobial activity. Inhibition of FPP synthase by N-BP may also mimic the endogenous antiviral effect of 25-hydroxycholesterol (25HC), one of the routes by which IFN signalling suppresses the flux through the mevalonate pathway.

the percentage or total number of AMΦ, or in the number or viability of cells in BAL fluid (*Figure 3—figure supplement 1*).

I.p. LPS challenge of ZOL-treated mice also resulted in a significant elevation in IL-1β in peritoneal fluid, and CCL2 and CCL5 in serum, compared to controls (*Figure 3c*). Furthermore, peritoneal cells from ZOL-treated mice produced almost three times more IL-1β upon ex vivo stimulation with LPS and nigericin (a well-described NLRP3 activator) (*Figure 3d*) than cells from control mice. IL-1β is primarily produced by monocytes/macrophages (*Kany et al., 2019*), the predominant cell type in PL (*Figure 1—figure supplement 1*). Thus, our results strongly suggest that macrophages are the most likely source of IL-1β in PL fluid. Importantly, treating the mice with ZOL did not cause IL-1β release from peritoneal cells stimulated ex vivo with LPS in the absence of nigericin (*Figure 3d*), and this is in accord with our previous finding that inhibition of the mevalonate pathway alone does not trigger NLRP3 inflammasome assembly but enhances its activation (*Skinner et al., 2019*).

The observations described here begin to provide a plausible mechanistic explanation for the decreased risk of pneumonia mortality associated with N-BP treatment (*Colón-Emeric et al., 2010*; *Sing et al., 2020*; *Reid et al., 2021*). AMΦ are one of the first lines of defence against common respiratory tract infections (*Byrne et al., 2015*), and inhibition of the mevalonate pathway in these cells may help boost the initial response to bacterial as well as viral lung infections by a variety of routes (*Figure 4*). First, uptake of N-BP into cells causes the accumulation of IPP/DMAPP (*Figure 2b*), which activates human Vγ9Vδ2-T cells (*Thompson and Rogers, 2004*). Vγ9Vδ2-T cells are non-conventional T cells with potent antibacterial and antiviral activity that recognise phosphoantigens, including IPP, derived from the mevalonate or DOXP pathways in bacterial pathogens (*Tanaka et al., 1995*; *Jomaa et al., 1999*). There is considerable interest in the use of N-BP-expanded γ,δ-T cells as an immunotherapy for cancer (*Clézardin and Massaia, 2010*; *Tanaka, 2020*) as well as viral diseases (*Juno and Kent, 2020*). Indeed, N-BP-expanded Vγ9Vδ2-T cells reduce disease severity and mortality from

influenza A virus (IAV) infection in humanised mice (*Tu et al., 2011*; *Zheng et al., 2015*). Second, the genome of some pathogens such as IAV encodes proteins with a prenylation motif, which require the host cells' mevalonate pathway to enable prenylation and allow pathogen propagation (*Marakasova et al., 2017*). Agents that block the mevalonate pathway (such as statins) or that inhibit prenylation (such as lonafarnib, currently in clinical trials for hepatitis delta virus infection), have well-described antiviral or antimicrobial effects (*Parihar et al., 2019*; *Einav and Glenn, 2003*). Intriguingly, simvastatin was shown to improve outcomes in hospitalised older adults with community-acquired pneumonia (*Sapey et al., 2019*). Third, inhibition of FPP synthase by N-BP in AMΦ mimics the decreased flux through the mevalonate pathway in macrophages in response to endogenous IFN signalling, which serves to limit viral uptake and replication by several mechanisms including the synthesis of 25-hydroxycholesterol (*Robertson and Ghazal, 2016*; *Cyster et al., 2014*). Fourth, lack of protein prenylation (*Figure 2c*) enhances NLRP3 inflammasome activation and promotes the release of IL-1β (*Skinner et al., 2019*). IL-1β is a central mediator of the innate immune response that orchestrates the production of a cascade of cytokines and chemokines (*Garlanda et al., 2013*). Our observation that systemic ZOL treatment significantly enhanced the release of IL-1β and several other cytokines and chemokines in lung, peritoneum, and serum after LPS challenge (*Figure 3*) is consistent with increased inflammasome activation. Our findings are also consistent with previous reports suggesting that ZOL treatment causes polarisation of tumour-associated macrophages towards an M1-like, pro-inflammatory phenotype by inhibiting the mevalonate pathway (*Coscia et al., 2010*; *Comito et al., 2017*). Further studies are necessary to determine exactly how ZOL alters the phenotype of AMΦ. Importantly however, despite the ability of ZOL to enhance cytokine and chemokine production, to our knowledge there is no evidence that N-BP therapy exacerbates lung inflammation in pneumonia patients and, on the contrary, N-BP treatment appears to have a beneficial effect on pneumonia risk and mortality (*Colón-Emeric et al., 2010*; *Sing et al., 2020*; *Reid et al., 2021*).

Finally, it is noteworthy that ZOL was recently identified, using computational biology approaches, as one of the 200 clinically approved drugs that are predicted to target pathways induced by SARS-CoV-2 and could be suitable for drug repurposing against COVID-19 (*Han et al., 2021*). Epidemiological studies are therefore urgently needed to determine whether N-BP therapy alters the incidence, severity, or risk of mortality from SARS-Cov-2 infection.

## Conclusions

We show that systemically administered N-BP drug can act outside the skeleton, inhibiting the mevalonate pathway and preventing protein prenylation in tissue-resident macrophages, which in turn enhances macrophage responsiveness to bacterial endotoxin. Our observations in mice, together with data from clinical studies in humans (*Colón-Emeric et al., 2010*; *Sing et al., 2020*; *Reid et al., 2021*), suggest that the beneficial effects of these drugs against pneumonia infection and mortality are, at least in part, mediated by targeting AMΦ, thereby boosting early immune responses in the lung. These findings add further weight to the view that N-BP therapy has benefits beyond just preventing bone loss and could be considered as prophylactic treatment to reduce the risk of pneumonia in individuals with osteopenia or osteoporosis, who are already eligible for bisphosphonate treatment under standard clinical guidelines.

## Materials and methods

### Key resources table

| Reagent type (species) or resource | Designation | Source or reference | Identifiers | Additional information |
|---|---|---|---|---|
| Antibody | BUV395 anti-CD11b (rat monoclonal) | BD Biosciences | Clone M1/70 | Flow cytometry (1:200) |
| Antibody | Biotin anti-F4/80 (rat monoclonal) | BioLegend | Clone BM8 | Flow cytometry (1:200) |
| Antibody | FITC anti- I-A/I-E (MHC-II) (rat monoclonal) | BD Biosciences | Clone 2G9 | Flow cytometry (1:200) |
| Antibody | BB515 anti-Siglec-F (rabbit monoclonal) | BD Biosciences | Clone E50-2440 | Flow cytometry (1:200) |

*Continued on next page*

*Continued*

| Reagent type (species) or resource | Designation | Source or reference | Identifiers | Additional information |
|---|---|---|---|---|
| Antibody | PE anti-CD11c (hamster monoclonal) | BD Biosciences | Clone N418 | Flow cytometry (1:200) |
| Antibody | PerCP-Cy5.5 anti-Ly6G (rat monoclonal) | BioLegend | Clone IA8 | Flow cytometry (1:200) |
| Antibody | BUV737 anti-B220 (rat monoclonal) | BD Biosciences | Clone RA3-6B2 | Flow cytometry (1:300) |
| Antibody | PE-Cy7 anti-Ly6C (rat monoclonal) | BD Biosciences | Clone AL-21 | Flow cytometry (1:200) |
| Antibody | APC-Cy7 anti- TCRb (hamster monoclonal) | BD Biosciences | Clone H57-597 | Flow cytometry (1:300) |
| Antibody | Anti-CD16/CD32 (rat monoclonal Fc block) | BD Biosciences | Clone 2.4G2 | Flow cytometry (1:200) |
| Peptide, recombinant protein | Streptavidin | BD Biosciences | BV421-streptavidin | Flow cytometry (1:400) |
| Chemical compound, drug | Zombie Aqua | BioLegend | Cat#: 423101 | Flow cytometry (1:700) |
| Chemical compound, drug | AF647-ZOL | BioVinc, CA | Cat#: SKU BV501001 | Flow cytometry (47.5 µg/40 nmoles per mouse i.v. dose) |
| Chemical compound, drug | Zoledronic acid | Sigma-Aldrich | Cat#: SML0223 | 500 µg/kg i.v. dose |
| Chemical compound, drug | LPS (*Escherichia coli*) | Sigma-Aldrich | O111:B4 | 10 µg i.n. or 100 µg i.p. per mouse |
| Chemical compound, drug | Biotin-GPP | doi.org.10.1080/21541248.2015.1085485 | | Prof Kirill Alexandrov |
| Peptide, recombinant protein | REP-1 (zebrafish) | doi.org.10.1080/21541248.2015.1085485 | | Prof Kirill Alexandrov |
| Peptide, recombinant protein | GGTase-II (rat) | doi.org.10.1080/21541248.2015.1085485 | | Prof Kirill Alexandrov |
| Peptide, recombinant protein | Streptavidin-680RD | LiCOR | P/N: 925-68079 | Western blotting (1:20,000) |
| Antibody | Anti-IL-1β (goat polyclonal) | R&D Systems | Cat#: AF-401-NA | Western blotting (1:1000) |
| Antibody | HRP anti-goat IgG (donkey polyclonal) | Thermo Fisher | Cat#: A15999 | Western blotting (1:5000) |
| Commercial assay or kit | SuperSignal West Pico substrate | Thermo Fisher | Cat#: 34580 | Western blotting |
| Commercial assay or kit | Bio-Plex immunoassay kit | Bio-Rad | Cat#: M60009RDPD | Cytokine and chemokine assay |
| Recombinant protein | M-CSF (human) | Sino Biological | Cat#: 11792-HNAH | Cell culture supplement (50 ng/mL) |
| Chemical compound | Nigericin | Sigma-Aldrich | Cat#: N7143 | |
| Chemical compound | IPP triammonium salt | Sigma-Aldrich | Cat#: I0503-1VL | LC-MS/MS standard |
| Chemical compound | DMAPP triammonium salt | Toronto Research Chemicals | Cat#: 63180-1MG | LC-MS/MS standard |
| Software, algorithm | FlowJo software | Becton Dickinson | RRID:SCR_008520 | Version 10.6.2 |
| Software, algorithm | MassHunter Quantitative Analysis software | Agilent | | Version B08.00.00 |

## Animals and tissue collection

Studies involving mice were performed in strict accordance with the Australian Code for the care and use of animals for scientific purposes (2013). All of the animals were handled according to Animal Ethics Committee protocols (Animal Research Authority: 18/40) approved by the Garvan Institute/ St Vincent's Hospital Animal Ethics Committee. Procedures were performed under appropriate anaesthesia, with animal welfare consideration underpinned by the principles of Replacement, Reduction, and Refinement.

All experiments involved adult female C57BL/6JAusb mice, and group sizes were based on previous studies using the same methodologies. Animals were purchased from Australian BioResources, housed with standard chow diet in specific pathogen-free conditions, and randomly allocated to experimental groups. Mice were anaesthetised with isoflurane prior to retro-orbital i.v., i.n., or i.p. drug administration. Mice were euthanised by $CO_2$ inhalation, and peritoneal and bronchoalveolar cells were isolated by lavage using 2 mM EDTA/magnesium- and calcium-free DPBS (Gibco). PL cells were collected by injecting 5 mL of solution into the peritoneal cavity, and BAL cells by pooling three consecutive 1.5 mL washes administered via an insertion in the trachea. To generate BMDM, the femora and tibiae of adult mice were flushed with sterile PBS. Bone marrow cells were cultured for 4 days in Petri dishes in RPMI supplemented with 10% heat-inactivated foetal calf serum, 50 units/mL penicillin; 50 µg/mL streptomycin (Gibco) and 50 ng/mL rhM-CSF (Sino Biological), in a humidified incubator with 5% $CO_2$ at 37°C.

## Flow cytometry

A single dose (47.5 µg /40 nmoles) of AF647-ZOL (BioVinc, CA) or 500 µg/kg ZOL (or saline vehicle) was administered i.v. in a final volume of 100 µL. Uptake of AF647-ZOL was assessed in BAL and PL cells 4 hr later. For LPS challenge, mice were treated with i.v. ZOL 48 hr prior to administration of i.n. LPS (10 µg LPS in 20 µL saline) or i.p. LPS (100 µg in 200 µL saline). Cell viability and total cell numbers in BAL and PL were assessed by trypan blue staining using a Corning CytoSMART cell counter. For flow cytometric analysis, cells were pre-incubated with mouse-Fc block and viability marker (Zombie Aqua viability stain, 1:700), prepared in calcium/magnesium-free PBS (Gibco), before staining with fluorescently conjugated antibodies prepared in staining buffer (2 mM EDTA, 0.02% azide; 0.5% foetal calf serum, in calcium/magnesium-free PBS). Antibodies and dilutions are listed in the Key resources table. Samples were analysed using a BD LSRII SORP flow cytometer/DIVA software. The post-acquisition analysis was performed using FlowJo 10.6.2 (BD).

## LC-MS/MS analysis of IPP/DMAPP

10-week-old female mice were administered a single 100 µL retro-orbital i.v. dose of 500 µg/kg ZOL (or saline control). Animals were culled 48 hr later, and bronchoalveolar cells and peritoneal cells were collected by pooling the BAL or PL lavages from n = 8 mice (BAL) or n = 5 mice (PL). Cell pellets were then stored at –80°C. For analysis of IPP/DMAPP, 1 mL cold extraction solvent (80:20 methanol:water) was added to the cell pellets, then vortexed for 10 s and incubated in a ultrasonic bath filled with ice water for 1 hr, then centrifuged at 3000 rpm for 30 min at 4°C. Aliquots (850 µL) of supernatant were dried under vacuum in an Eppendorf Concentrator Plus then reconstituted in 42.5 µL 70% methanol, 30% 10 mM ammonium acetate. Samples (injection volume 1 µL) were analysed by targeted LC-MS/MS using an Agilent 1290 Infinity II UHPLC system coupled to an Agilent 6495B triple quadrupole mass spectrometer. Separation was achieved using an Agilent Infinity Poroshell 120 EC-C18 column (3.0 × 150 mm, 2.7 µm) fitted with an Agilent Infinity Poroshell 120 EC-C18 UHPLC guard column (3.0 × 150 mm, 2.7 µm), maintained at 20°C. The mobile phases were 10 mM ammonium acetate in water (A) and methanol (B), both containing 5 µM medronic acid (Sigma-Aldrich) to chelate metal ions (gradient 98% A from 0 to 3 min, decreased to 2% A from 3.5 to 6.5 min at 0.5 mL/min, then increased to 98% A at 0.4 mL/min from 6.5 to 12 minutes; total run time 12 min). Autosampler temperature was 4°C. The mass spectrometer was operated in negative electrospray ionisation mode: source gas temperature was 250°C with flow at 17 L/min, sheath gas temperature was 400°C with flow at 12 L/min, and nebuliser pressure was 45 psi. Data were acquired in Multiple Reaction Monitoring (MRM) mode and were processed using Agilent MassHunter Quantitative Analysis software version B08.00.00. By comparison with pure standard compounds (Sigma-Aldrich), the isomers IPP and DMAPP eluted at the same retention time (approximately 2.4 min) and were calculated as total area under the curve. The limit of detection in cell extracts was 20 nM.

## Detection of unprenylated Rab proteins

To assess the accumulation of unprenylated Rab GTPase proteins, we used an in vitro prenylation assay as previously described (*Ali et al., 2015*). Mice were treated with i.v. ZOL or saline as described above, then cell pellets were obtained by BAL or PL (each pooled from n = 5 mice) and lysed by sonication in prenylation buffer (50 mM HEPES, pH 7.2, 50 mM NaCl, 2 mM $MgCl_2$, 100 µM GDP, 1× Roche

cOmplete EDTA-free protease inhibitor cocktail). To compare the effect of ZOL and AF647-ZOL, BMDMs were treated with 1, 5, or 10 µM ZOL or AF647-ZOL for 24 hr before cells were collected and lysed in prenylation buffer.

For the in vitro prenylation assay, 10 µg of protein were incubated with recombinant GGTase II, REP-1, and biotin-conjugated GPP (a synthetic isoprenoid lipid) for the labelling of unprenylated Rab proteins (*Ali et al., 2015*). The resulting biotinylated Rabs were then detected on PVDF blots using streptavidin-680RD (LiCOR). A narrow doublet (often appearing as a broad singlet) of endogenous biotinylated 75 kDa proteins was used as a sample loading control.

### Immune responses to LPS in vivo

10-week-old female mice were administered a single retro-orbital i.v. dose of 500 µg/kg ZOL (or saline control), 48 hr before immune challenge with LPS (*Escherichia coli* O111:B4, Sigma-Aldrich) administered either i.n. (10 µg LPS in 20 µL saline) or i.p. (100 µg in 200 µL saline). Mice were culled 2 hr later, and BAL or PL fluid were collected by injecting 500 µL or 1 mL PBS into the lungs or peritoneal cavity, respectively. Cytokines and chemokines in BAL, PL, and serum were measured using a Bio-Plex multiplex immunoassay (Bio-Rad) and a MAGPIX (Luminex) multiplex reader according to the manufacturer's instructions.

### IL-1β release by peritoneal cells ex vivo

PL samples were obtained from ZOL- or saline-treated mice as described above. PL cells were placed in 96-well plates (400,000 cells/well) and treated at 37°C for 5.25 hr with 200 ng/mL LPS, in a final volume of 200 µL serum-free Opti-MEM medium (Gibco), followed by stimulation for 45 min with 10 µM nigericin (Sigma-Aldrich). The level of mature (17 kDa) IL-1β in conditioned medium was analysed by western blotting on nitrocellulose membrane with a goat anti-mouse IL-1β antibody (AF-401-NA, R&D Systems, 1:1000 dilution), donkey anti-goat horseradish peroxidase-conjugated secondary antibody (A15999, Thermo Fisher, 1:5,000 dilution), and enhanced chemiluminescence using SuperSignal West Pico chemiluminescent substrate (Thermo Fisher). The signal was detected using a Fusion FX7 imaging system (Etablissements Vilber Lourmat SAS), and densitometry was performed on blots using ImageJ (v2.0.0).

## Acknowledgements

We thank Prof Kirill Alexandrov (Queensland University of Technology) and Dr Zakir Tnimov (MRC Laboratory of Molecular Biology) for providing reagents for the Rab prenylation assay. This work was supported in part by the National Health and Medical Research Council (NHMRC) of Australia project grant 1079522 to MJR, by Mrs Janice Gibson and the Ernest Heine Family Foundation, and a Perpetual IMPACT grant to MAM. EKF was supported through an Australian Government Research Training Program Scholarship. PMH was funded by a Fellowship and grants from the NHMRC (1175134) and by the University of Technology Sydney (UTS). We gratefully acknowledge funding by the New South Wales Government for the Victor Chang Cardiac Research Institute Innovation Centre, as well as funding from the Freedman Foundation for the Metabolomics Facility.

## Additional information

### Competing interests

Marcia A Munoz, Shuting Sun, Frank H Ebetino, Jacqueline R Center: FHE is an employee of BioVinc (Pasadena, CA, USA). SS also co-affiliates with BioVinc. JRC participates on advisory boards of Amgen and Bayer and receives payment from Amgen for educational activities. The other authors declare that no competing interests exist.

### Funding

| Funder | Grant reference number | Author |
|--------|------------------------|--------|
| NHMRC | 1079522 | Michael J Rogers |

| Funder | Grant reference number | Author |
|---|---|---|
| Perpetual IMPACT | | Marcia A Munoz |
| NHMRC | 1175134 | Philip M Hansbro |
| Australian Government Research Training Program Scholarship | | Emma K Fletcher |
| Mrs Janice Gibson and the Ernest Heine Family Foundation | | Michael J Rogers |

The funders had no role in study design, data collection and interpretation, or the decision to submit the work for publication.

## Author contributions

Marcia A Munoz, Conceptualization, Funding acquisition, Investigation, Project administration, Writing – review and editing; Emma K Fletcher, Oliver P Skinner, Julie Jurczyluk, Esther Kristianto, Mark P Hodson, Investigation, Methodology; Shuting Sun, Frank H Ebetino, David R Croucher, Methodology; Philip M Hansbro, Methodology, Writing – review and editing; Jacqueline R Center, Writing – review and editing; Michael J Rogers, Conceptualization, Funding acquisition, Project administration, Supervision, Writing – original draft

## Author ORCIDs

Marcia A Munoz ⓘ http://orcid.org/0000-0001-7603-0351
Mark P Hodson ⓘ http://orcid.org/0000-0002-5436-1886
David R Croucher ⓘ http://orcid.org/0000-0003-4965-8674
Jacqueline R Center ⓘ http://orcid.org/0000-0002-5278-4527
Michael J Rogers ⓘ http://orcid.org/0000-0002-1818-9249

## Ethics

Studies involving mice were performed in strict accordance with the Australian Code for the care and use of animals for scientific purposes (2013). All of the animals were handled according to Animal Ethics Committee protocols (Animal Research Authority: 18/40) approved by the Garvan Institute/ St Vincent's Hospital Animal Ethics Committee. Procedures were performed under appropriate anaesthesia, with animal welfare consideration underpinned by the principles of Replacement, Reduction and Refinement.

## Decision letter and Author response

Decision letter https://doi.org/10.7554/eLife.72430.sa1
Author response https://doi.org/10.7554/eLife.72430.sa2

---

# Additional files

## Supplementary files

• Transparent reporting form

## Data availability

All data generated or analysed during this study are included in the manuscript and measurements obtained from individual mice are shown as separate data points in the figures. Source data files are provided for the protein blots shown in Figure 1a, Figure 2c and Figure 3d.

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
