## [Editor Report]

Your paper is a careful analysis to understand the effects of bisphosphonates, long thought to act only on osteoclasts to block bone resorption, on the lung. You show that drugs in this class act on pulmonary macrophages and block protein prenylation. This leads to an enhanced response to bacterial endotoxin and immune response to bacterial endotoxin with increased levels of cytokines and chemokines in bronchoalveolar fluid – a series of events that may explain reduced rates of pneumonia seen in patients treated with these drugs. This clinical observation has long defied our understanding.

---

## [Decision Letter]

**Decision letter after peer review:**

Thank you for submitting your article "Bisphosphonate drugs have actions outside the skeleton and inhibit the mevalonate pathway in alveolar macrophages" for consideration by *eLife*. Your article has been reviewed by 3 peer reviewers, and the evaluation has been overseen by a Reviewing Editor and Carlos Isales as the Senior Editor. The following individuals involved in review of your submission have agreed to reveal their identity: T. Jack Martin (Reviewer #1); Massimo Massaia (Reviewer #2); Lilian Plotkin (Reviewer #3).

Essential revisions:

The Reviewing Editor believes that there are very few time-consuming requests in the 3 reviews above, unless I am incorrect. I believe the authors can respond to each Reviewer in a succinct manner as they are enthusiastic overall regarding the data, the paper and the topic.

*Reviewer #1 (Recommendations for the authors):*

1. Figure 1. AF647ZOL after IV injection in the mouse was clearly localized in pulmonary alveolar and in peritoneal macrophages. AF647-ZOL is a fluorescent-labelled zoledronate. The authors should provide more details in Methods about this reagent, its preparation and properties. There is nothing published about it of which I am aware. The literature on the corresponding risedronate analog, AF647-RIS, is confusing. It was used in Junankar et al., where it was described as bone-seeking but pharmacologically inactive , but this appears to differ from the original description, in Roelofs et al., (JBMR 2010 ). Perhaps the authors could clarify.

2. The AF647-ZOL was used only in the work of Figure 1, where the localization in macrophages is clearly shown. A couple of questions arise. First, if the labelled compound is active, sufficient was injected in Figure 1 to show a pharmacological effect as was studied in Figure 2 – was there any biological activity of AF647ZOL – or a reduced activity?? Second, it is assumed, but not shown, that the labelled ZOL behaves as unlabelled. As one would do with any binding or "receptor" study, one would see if it were possible to block the label from entering cells. This could be done, for example, by examining mice co-injected with an increased amount of unlabelled ZOL. The question is being asked to increase confidence that the pharmacological effects shown in Figures2 and 3 are indeed the result of direct action of the unlabelled ZOL.

3. The present experiments were carried out with one N-BP, the most potent of all those in common use. Much higher doses of that drug are required in mice than in human subjects, but it would be very helpful to know what is needed to produce the pharmacologic effects that are shown. The impracticality of of dose responses is obvious, but what about repeating the Figure 2 experiment with less active N-BPs/ – e.g. risedronate, alendronate, pamidronate?

4. Given that ZOL is active on bone in human studies when given with as little as 1 microgram intermittent doses, the authors would do well to discuss how this question of bioavailability is to be addressed. It might well be beyond the scope of this paper, but it surely should be discussed.

*Reviewer #2 (Recommendations for the authors):*

1. I would recommend a more cautious approach before stating that "these findings demonstrate unequivocally for the first time that systemic administration of N-BP has pharmacological activity outside the skeleton" (page 5, line 10, and thereafter). Similar results have already been reported by several groups (Rogers TL et al., Macrophages as potential targets for zoledronic acid outside the skeleton-evidence from in vitro and in vivo models. Cell Oncol (Dordr). 2013 Dec;36(6):505-14; Comito G et al., Zoledronic acid impairs stromal reactivity by inhibiting M2-macrophages polarization and prostate cancer-associated fibroblasts. Oncotarget. 2017 Jan 3;8(1):118-132; Coscia et al., J Cell Mol Med. 2010 Dec;14(12):2803-15). The paper by Coscia et al., was the first one to demonstrate that ZOL at clinically achievable doses targets the mevalonate pathway of tumor-associated macrophages (TAM) and tumor cells in a mouse model of primary tumor not located in the bone. In the same paper, it was shown a marked inhibition of the Mev pathway of peritoneal macrophages after ZOL treatment, as shown by the intracellular decrease of FPP, cholesterol, and ubiquinone, and the concurrent increase of IPP. As expected, protein prenylation was also affected as shown by the decreased intracellular contents of prenylated Ras and Ras-GTP. The Authors should reconsider and discuss again their results in light of these published data.

2. Interestingly, the uptake of the fluorescently-labelled ZOL analog (AF647-ZOL) is mostly restricted to large peritoneal (LP) macrophasegs sparing small peritoneal (SP) macrophages. The Authors should explain the functional difference between these 2 subsets and provide some hypotheses why uptake is restricted to the former (i.e, activated macrophages?). Is AF647-ZOL uptake specific or is this the consequence of a general increased phagocytic activity of LP macrophages (control expts with FITC-dextran could answer the question).

3. Another interesting set of data is represented by the enhanced cytokines and chemokines production induced by LPS after in vivo ZOL priming. These results confirm previous findings (Coscia et al., J Cell Mol Med. 2010 Dec;14(12):2803-15) that ZOL repolarizes TAM from M2 (protumoral) to M1 (antitumoral) activity by inhibiting intracellular protein prenylation. iNOS is considered a hallmark of M1 polarized macrophages and an important player of the M1-mediated tumoricidal activity. Do the Authors believe that some sort of M1 polarization and iNOS induction can play a role in the improved capacity of ZOL-primed to resist to LPS challenge and survive pneumonia?

4. As reported by the Authors, clinical data associating improved pneumonia outcome with ZOL treatment have been generated in aged patients. Have the Authors any evidence that age can influence the ability to prime alveolar or peritoneal macrophages in C57BL/6J mice in relation to age? Do the Authors have any explanation about a possible selective effect of ZOL in the elderly, or the possible association is simply because ZOL is used in the elderly to treat hip fractures or osteoporosis? Is any evidence about a protective effect of ZOL in young cancer patients treated with ZOL for bone disease (i.e., myeloma, breast cancer, prostate cancer, etc)?

5. It is very exciting that ZOL has been identified, using computational biology approaches, as one of 200 clinically-approved drugs that are predicted to target pathways induced by SARS-CoV-2 and could be suitable for drug repurposing against COVID-19 (page 7, line 11 onwards). However, one major cause of Sars-CoV-2 induced morbidity and mortality is the exaggerated inflammatory response in the lung. How do the Authors reconcile a protective effect against SARS-CoV-2 infection with the capacity of ZOL to boosts the reactivity of alveolar macrophages? Is LPS challenge a suitable surrogate to mimic viral infection? My personal feeling is that the concurrent activation of Vg9Vd2 T cells can be more important in viral infections. The Authors should comment on a possible difference between ZOL and statins in regulating inflammatory and immune responses against bacteria and viruses. I am also wondering whether statins (that induce and accumulate unprenylated small GTPase proteins like ZOL, but do not activate Vg9Vd2 T cells) have the same ability to prime alveolar macrophages reactivity to LPS challenge.

*Reviewer #3 (Recommendations for the authors):*

The manuscript is clearly written, and the data are properly presented. While the conclusions and potential relevance of the studies are clear and appropriate, the authors should also discuss the potential implications of their finding on the acute phase response observed in patients receiving bisphosphonates. Further, the potential of eliciting an exacerbated response that could be deleterious to the patients should also be noted.

---

## [Author Response]

Reviewer #1 (Recommendations for the authors):1. Figure 1. AF647ZOL after IV injection in the mouse was clearly localized in pulmonary alveolar and in peritoneal macrophages. AF647-ZOL is a fluorescent-labelled zoledronate. The authors should provide more details in Methods about this reagent, its preparation and properties. There is nothing published about it of which I am aware. The literature on the corresponding risedronate analog, AF647-RIS, is confusing. It was used in Junankar et al., where it was described as bone-seeking but pharmacologically inactive , but this appears to differ from the original description, in Roelofs et al., (JBMR 2010 ). Perhaps the authors could clarify.

We appreciate the Reviewer raising these questions, and realise that AF647-ZOL does indeed lack description in the literature. It is a commercially-available reagent and was sourced from BioVinc (as stated in the Methods). We now include an additional panel (a) in Figure 1 showing that AF647-ZOL, like AF647-RIS (described in Roelofs *et al.,* 2010) does not inhibit protein prenylation in cultured macrophages at equimolar concentrations to unlabelled ZOL. AF647-ZOL is therefore a valuable tool to examine the distribution of bisphosphonate compounds in vivo but lacks pharmacologic activity. This is now made clear in the revised text (1^st^ paragraph of Results and Discussion). To clarify, the study by Roelofs *et al.,* (JBMR 2010) demonstrates that FAM-RIS is pharmacologically active but clearly states that, unlike FAM-RIS, “AF647-RIS did not inhibit Rap1A prenylation in J774.2 cells in vitro”.

2. The AF647-ZOL was used only in the work of Figure 1, where the localization in macrophages is clearly shown. A couple of questions arise. First, if the labelled compound is active, sufficient was injected in Figure 1 to show a pharmacological effect as was studied in Figure 2 – was there any biological activity of AF647ZOL – or a reduced activity?? Second, it is assumed, but not shown, that the labelled ZOL behaves as unlabelled. As one would do with any binding or "receptor" study, one would see if it were possible to block the label from entering cells. This could be done, for example, by examining mice co-injected with an increased amount of unlabelled ZOL. The question is being asked to increase confidence that the pharmacological effects shown in Figures2 and 3 are indeed the result of direct action of the unlabelled ZOL.

As described above (and in the revised text and Figure 1a), we now explain that AF647-ZOL is pharmacologically inactive and was only used in Figure 1 to reveal cellular uptake by macrophages in vivo. In Figures 2 and 3, only unlabelled ZOL was used. Given that ZOL is a well-described, potent bisphosphonate inhibitor of the mevalonate pathway, and that the only cells in BAL and PL samples that internalised bisphosphonate were macrophages, we cannot think of any other explanation for the results in Figure 2 other than a direct effect of ZOL on the mevalonate pathway in AMφ and PMφ. We do not suggest that all of the cytokines and chemokines measured in Figure 3 are derived solely from macrophages, and the exact source and mechanism for the increase remains to be determined. Single cell RNA sequencing experiments are underway in a follow-up study to examine this in more detail but we feel that this is beyond the scope of the current Short Report. Nevertheless, intranasal or intraperitoneal administration of LPS is an established method for activating AMφ and PMφ and inducing IL-1β release from these cells. We state (page 7) that: “IL-1β is a central mediator of the innate immune response that orchestrates the production of a cascade of cytokines and chemokines (Garlanda et al., 2013)” therefore it is reasonable to expect that at least some of the acute increase in IL1-β and other cytokine/chemokines in BAL fluid and PL fluid occurs directly from AMφ and PMφ respectively. Furthermore, cytokine/chemokine release in BAL and PL fluid was measured just 2 hours after LPS administration, before infiltration by other immune cells occurs.

3. The present experiments were carried out with one N-BP, the most potent of all those in common use. Much higher doses of that drug are required in mice than in human subjects, but it would be very helpful to know what is needed to produce the pharmacologic effects that are shown. The impracticality of of dose responses is obvious, but what about repeating the Figure 2 experiment with less active N-BPs/ – e.g. risedronate, alendronate, pamidronate?

The single dose of ZOL that was used in the studies in Figures 2 and 3 is reasonably comparable (in terms of µg ZOL/kg body weight) to the clinical dose used in humans. Because of the bone-targeting property of bisphosphonates such as ZOL the exact concentration that is present in soft tissues such as lung is extremely low (see Green and Rogers, Drug Dev Res 55:210–224, 2002). As we suggest in the text (page 3), the ability of macrophages to internalise bisphosphonate in tissues outside the skeleton presumably reflects their enormous capacity for rapid endocytosis. We agree that it will be important to assess the ability of orally-administered bisphosphonates to have effects on AMφ, but we respectfully consider this to be beyond the scope of this initial Short Report.

4. Given that ZOL is active on bone in human studies when given with as little as 1 microgram intermittent doses, the authors would do well to discuss how this question of bioavailability is to be addressed. It might well be beyond the scope of this paper, but it surely should be discussed.

With respect, we are unsure what the Reviewer is asking. Despite the rapid bone-targeting property of bisphosphonates it has been shown in rodents that small amounts of ZOL can accumulate in soft tissues (Green and Rogers, *Drug Dev Res* 55:210–224, 2002) but the concentrations that can be achieved are unknown. This has now been stated in the text (page 4). The randomised clinical trial by Grey *et al.,* (*J Bone Miner Res* 2014) showed that ZOL is pharmacologically active in humans when administered intravenously at intermittent doses of 1 milligram (not 1 microgram). In our study in mice we used a single dose of 500 micrograms/kg, which is equivalent to about 10 micrograms per mouse, and was delivered intravenously (as in humans). As discussed in response to point 3 above, it will be important to determine in future studies whether orally-administered N-BP is sufficiently bioavailable to affect tissue-resident macrophage populations such as AMφ, but this is beyond the scope of the current study in which we clearly show an effect of a single dose of intravenous ZOL.

Reviewer #2 (Recommendations for the authors):1. I would recommend a more cautious approach before stating that "these findings demonstrate unequivocally for the first time that systemic administration of N-BP has pharmacological activity outside the skeleton" (page 5, line 10, and thereafter). Similar results have already been reported by several groups (Rogers TL et al., Macrophages as potential targets for zoledronic acid outside the skeleton-evidence from in vitro and in vivo models. Cell Oncol (Dordr). 2013 Dec;36(6):505-14; Comito G et al., Zoledronic acid impairs stromal reactivity by inhibiting M2-macrophages polarization and prostate cancer-associated fibroblasts. Oncotarget. 2017 Jan 3;8(1):118-132; Coscia et al., J Cell Mol Med. 2010 Dec;14(12):2803-15). The paper by Coscia et al., was the first one to demonstrate that ZOL at clinically achievable doses targets the mevalonate pathway of tumor-associated macrophages (TAM) and tumor cells in a mouse model of primary tumor not located in the bone. In the same paper, it was shown a marked inhibition of the Mev pathway of peritoneal macrophages after ZOL treatment, as shown by the intracellular decrease of FPP, cholesterol, and ubiquinone, and the concurrent increase of IPP. As expected, protein prenylation was also affected as shown by the decreased intracellular contents of prenylated Ras and Ras-GTP. The Authors should reconsider and discuss again their results in light of these published data.

We appreciate the Reviewer’s comment. Our intention was to make the point that a single dose of ZOL can affect cells outside the skeleton under benign conditions i.e. not in mice bearing tumours and with potentially tumour-influenced immune systems, but we realise that this was not clear. We propose to rephrase the title, from “Bisphosphonate drugs have actions outside the skeleton and inhibit the mevalonate pathway in alveolar macrophages” to “Bisphosphonate drugs have actions in the lung and inhibit the mevalonate pathway in alveolar macrophages”. This more specifically reflects the novel findings that we present in the Short Report. As suggested, we have also modified the sentence on page 5 and added an additional short section toward the end of the Results and Discussion (page 7) to acknowledge previous findings in tumour-bearing mice.

2. Interestingly, the uptake of the fluorescently-labelled ZOL analog (AF647-ZOL) is mostly restricted to large peritoneal (LP) macrophasegs sparing small peritoneal (SP) macrophages. The Authors should explain the functional difference between these 2 subsets and provide some hypotheses why uptake is restricted to the former (i.e, activated macrophages?). Is AF647-ZOL uptake specific or is this the consequence of a general increased phagocytic activity of LP macrophages (control expts with FITC-dextran could answer the question).

This is indeed an interesting question. An explanation for the specific uptake of bisphosphonate by large PMφ compared to small PMφ requires further detailed studies (both subsets are capable of phagocytosis and endocytosis). Since our studies were more focused on AMφ rather than PMφ we consider this to be beyond the scope of this Short Report. However, we now indicate (page 4) that these two PMφ subsets differ in their origin, phenotype and function, which is explained in detail in the cited review.

3. Another interesting set of data is represented by the enhanced cytokines and chemokines production induced by LPS after in vivo ZOL priming. These results confirm previous findings (Coscia et al., J Cell Mol Med. 2010 Dec;14(12):2803-15) that ZOL repolarizes TAM from M2 (protumoral) to M1 (antitumoral) activity by inhibiting intracellular protein prenylation. iNOS is considered a hallmark of M1 polarized macrophages and an important player of the M1-mediated tumoricidal activity. Do the Authors believe that some sort of M1 polarization and iNOS induction can play a role in the improved capacity of ZOL-primed to resist to LPS challenge and survive pneumonia?

We take a more cautious approach and have now stated in the revised text that:

“Our findings are also consistent with previous reports suggesting that ZOL treatment causes polarisation of tumour-associated macrophages towards an M1-like, pro-inflammatory phenotype….”

We do not wish to state that our findings *confirm* this suggestion because this requires detailed analysis of the phenotype of AMφ and PMφ. As indicated to Reviewer #1, single cell RNA sequencing studies (not just iNOS expression) are underway to address this question in far more detail and to provide a complete explanation for the effect of ZOL, but we feel that this is beyond the scope of the current Short Report.

4. As reported by the Authors, clinical data associating improved pneumonia outcome with ZOL treatment have been generated in aged patients. Have the Authors any evidence that age can influence the ability to prime alveolar or peritoneal macrophages in C57BL/6J mice in relation to age? Do the Authors have any explanation about a possible selective effect of ZOL in the elderly, or the possible association is simply because ZOL is used in the elderly to treat hip fractures or osteoporosis? Is any evidence about a protective effect of ZOL in young cancer patients treated with ZOL for bone disease (i.e., myeloma, breast cancer, prostate cancer, etc)?

We are not aware of any studies to suggest that there is a selective effect of ZOL in the elderly and this could be a topic to address in future work. Of course, the elderly population are the most likely to receive bisphosphonate therapy for age-associated bone disorders such as osteoporosis, and are also more susceptible to pulmonary infections and pneumonia. We have now rephrased the last sentence of the Conclusion to make the clinical significance of our findings clearer:

“These findings add further weight to the view that N-BP therapy has benefits beyond just preventing bone loss and could be considered as prophylactic treatment to reduce the risk of pneumonia in individuals with osteopenia or osteoporosis, who are already eligible for bisphosphonate treatment under standard clinical guidelines”.

5. It is very exciting that ZOL has been identified, using computational biology approaches, as one of 200 clinically-approved drugs that are predicted to target pathways induced by SARS-CoV-2 and could be suitable for drug repurposing against COVID-19 (page 7, line 11 onwards). However, one major cause of Sars-CoV-2 induced morbidity and mortality is the exaggerated inflammatory response in the lung. How do the Authors reconcile a protective effect against SARS-CoV-2 infection with the capacity of ZOL to boosts the reactivity of alveolar macrophages? Is LPS challenge a suitable surrogate to mimic viral infection? My personal feeling is that the concurrent activation of Vg9Vd2 T cells can be more important in viral infections. The Authors should comment on a possible difference between ZOL and statins in regulating inflammatory and immune responses against bacteria and viruses. I am also wondering whether statins (that induce and accumulate unprenylated small GTPase proteins like ZOL, but do not activate Vg9Vd2 T cells) have the same ability to prime alveolar macrophages reactivity to LPS challenge.

It is worth re-emphasising that ZOL is only *predicted* to target pathways important for SARS-CoV-2 infection and there is no evidence yet, to our knowledge, that ZOL has any clinical benefit against COVID-19. Whether there is any clinical benefit, or indeed a deleterious effect, of ZOL therapy remains to be determined. We state (towards the end of the Results and Discussion) that there is evidence from epidemiological studies for only a *beneficial* effect of bisphosphonates on pneumonia and no evidence for a *deleterious* effect, despite the apparent pro-inflammatory actions of ZOL. The possible mechanisms underlying the beneficial effect on pneumonia are described in the text and in Figure 4. It is important to bear in mind that a rapid response to pathogens may lead to a better resolution of the infection. Hence an improved acute and early immune response in the lung may lead to overall better outcomes. The findings that we present in this Short Report reveal a possible explanation for the apparent beneficial effects of bisphosphonates against pneumonia and open a whole new area of research – the questions raised by the Reviewer remain to be explored. We do not suggest that LPS administration is a model of viral infection and we cite the studies that have previously reported anti-viral effects of ZOL. We totally agree that activation of Vγ9Vծ2 T cells is likely to be a major component of any effect of bisphosphonates to combat infection (both viral and bacterial) and discuss this in the text and in Figure 4, but naturally this cannot be examined in normal mice because they lack this subset of γ,ծ-T cells. Whether statins, or other pharmacologic agents that are capable of inhibiting steps of the mevalonate pathway, can also prime alveolar macrophages in vivo remains to be determined and is beyond the scope of this Short Report, but we cite key references that describe the effects of statins on infection (e.g. Parihar et al., 2019; Sapey et al., 2019).

Reviewer #3 (Recommendations for the authors):The manuscript is clearly written, and the data are properly presented. While the conclusions and potential relevance of the studies are clear and appropriate, the authors should also discuss the potential implications of their finding on the acute phase response observed in patients receiving bisphosphonates. Further, the potential of eliciting an exacerbated response that could be deleterious to the patients should also be noted.

We thank the Reviewer for this comment and to address it we have added a sentence (page 5) that states:

“Our findings also shed new light on the commonest side-effect of N-BP therapy, a transient acute phase response. This was previously thought to be caused by the accumulation of IPP in circulating monocytes, which then activates Vγ9Vծ2 cells to produce TNFα and IFNγ (Roelofs et al., 2009). Our findings suggest that tissue-resident macrophages, such as AMφ and large PMφ, are in fact the more likely source of IPP responsible for the activation of Vγ9Vծ2 cells”.

As described in the response to Reviewer #2, we already state that there is no evidence that enhanced cytokine production caused by N-BP therapy worsens lung inflammation in pneumonia patients, and on the contrary, N-BP treatment appears to have a beneficial effect on pneumonia risk and mortality. We have now emphasised this further in the revised text (page 7) by stating:

“Further studies are necessary to determine exactly how ZOL alters the phenotype of AMφ. Importantly however, despite the ability of ZOL to enhance cytokine and chemokine production, to our knowledge there is no evidence that N-BP therapy exacerbates lung inflammation in pneumonia patients, and on the contrary, treatment appears to have a beneficial effect on pneumonia risk and mortality (Colon-Emeric et al., 2010, Sing et al., 2020, Reid et al., 2021)”.